# αV Integrin Expression and Localization in Male Germ Cells

**DOI:** 10.3390/ijms22179525

**Published:** 2021-09-02

**Authors:** Veronika Palenikova, Michaela Frolikova, Eliska Valaskova, Pavla Postlerova, Katerina Komrskova

**Affiliations:** 1Laboratory of Reproductive Biology, Institute of Biotechnology of the Czech Academy of Sciences, BIOCEV, Prumyslova 595, 252 50 Vestec, Czech Republic; veronika.palenikova@ibt.cas.cz (V.P.); michaela.frolikova@ibt.cas.cz (M.F.); eliska.valaskova@ibt.cas.cz (E.V.); pavla.postlerova@ibt.cas.cz (P.P.); 2Department of Biochemistry, Faculty of Science, Charles University, Hlavova 8, 128 40 Prague 2, Czech Republic; 3Department of Veterinary Sciences, Faculty of Agrobiology, Food and Natural Resources, University of Life Sciences Prague, Kamycka 129, 165 00 Prague 6, Czech Republic; 4Department of Zoology, Faculty of Science, Charles University, BIOCEV, Vinicna 7, 128 44 Prague 2, Czech Republic

**Keywords:** αV integrin, male germ cells, sperm, mouse, pig, human

## Abstract

Integrins are transmembrane receptors that facilitate cell adhesion and cell–extracellular matrix communication. They are involved in the sperm maturation including capacitation and gamete interaction, resulting in successful fertilization. αV integrin belongs to the integrin glycoprotein superfamily, and it is indispensable for physiological spermiogenesis and testosterone production. We targeted the gene and protein expression of the αV integrin subunit and described its membrane localization in sperm. Firstly, in mouse, we traced *αV integrin* gene expression during spermatogenesis in testicular fraction separated by elutriation, and we detected gene activity in spermatogonia, spermatocytes, and round spermatids. Secondly, we specified αV integrin membrane localization in acrosome-intact and acrosome-reacted sperm and compared its pattern between mouse, pig, and human. Using immunodetection and structured illumination microscopy (SIM), the αV integrin localization was confined to the plasma membrane covering the acrosomal cap area and also to the inner acrosomal membrane of acrosome-intact sperm of all selected species. During the acrosome reaction, which was induced on capacitated sperm, the αV integrin relocated and was detected over the whole sperm head. Knowledge of the integrin pattern in mature sperm prepares the ground for further investigation into the pathologies and related fertility issues in human medicine and veterinary science.

## 1. Introduction

Integrin αV (also called CD51) belongs to the heterodimeric transmembrane family of integrins, which are crucial receptors for cell–cell adhesion and also cell adhesion to extracellular matrix proteins, and with their ligands play an irreplaceable role in successful fertilization and embryo development [1]. Integrin receptors can transduce the signal through the cell membrane in response to ligand binding, resulting in the regulation of intracellular pH via available free calcium ions or protein phosphorylation [2,3,4]. All these reactions are indispensable for sperm maturation and fertilization. Moreover, integrins associate with other membrane receptors and cytoskeletal proteins to create a multimolecular complex that participates in cell activation [3]. Within the oocyte tetraspanin web, integrins associate with members of the tetraspanin protein family and mediate mutual associations [5,6]. An analogous system is predicted to occur with sperm and CD9, CD81 [7,8], and CD151 [9]. αV integrin is also proposed to be involved in the formation of a multimolecular network [7,8,9,10]. The αV integrin recognizes ligands with the arginine–glycine–aspartic acid (RGD) sequence, resulting in the recognition of fibronectin, vitronectin, or osteopontin [11,12], which are present in the reproductive tract and can affect the interaction between the sperm and oocyte [10]. In the same way as other α integrin subunits, the αV integrin can be associated with β integrin subunits, and the αV/β3 integrin heterodimer has been identified on the mouse sperm membrane after capacitation and acrosome reaction [10]. This heterodimer was detected on the flagellum of mouse spermatozoa and could play a role in the fusion of oviductosomes that transport transmembrane proteins into the sperm [13]. The αV integrin was found on human acrosome-reacted spermatozoa, and based on empirical data, it was suggested to be confined to the inner acrosomal membrane [14].

The goal of our study was to identify the precise localization of the αV integrin subunit in intact sperm heads and to investigate its potential changes in membrane reorganization during the acrosome reaction initiated in capacitated sperm. The study targeted αV integrin membrane compartment localization in three species represented by two experimental models with distinct sperm head morphology such as mouse and pig and to compare them with human. A detailed understanding of αV integrin behavior prior to fertilization prepares the ground for future research involving sperm pathologies for the possibility of novel fertility-related protein markers.

## 2. Results

Owing to the dynamic expression of different integrins in germ cells and the importance of αV integrin during fertilization, we aimed to investigate its mRNA expression during spermatogenesis using the mouse model and then further describe the precise localization of αV integrin on sperm. To draw broader conclusions, we targeted αV integrin protein expression in individual sperm head membranes and followed its dynamic relocalization during sperm capacitation and acrosome reaction, in mouse, pig, and human. To achieve our aims, we used a wide range of methods such as elutriation of testicular suspension; qRT-PCR for *αV integrin* gene expression during spermatogenesis; immunofluorescent protein depiction; super-resolution microscopy, and Western blot detection in sperm and reproductive tissue extracts (Figure 1).

### 2.1. Mouse Sperm

To determine the expression of *αV integrin* in mouse, we prepared testicular suspensions and separated individual cell populations by elutriation [15,16]. Using individual cell fractions, we performed qRT-PCR to define cell type enrichment in the obtained fractions. We selected fractions enriched by male germ cell subtypes using specific gene markers for spermatogonia, primary spermatocytes, and round spermatids, which are summarized in Appendix A. In addition, we also defined the somatic cell-enriched fraction containing mainly Leydig cells. In identified fractions, we detected *αV integrin* mRNA expression in all five individual germ cell subtypes with particular enrichment in spermatogonia, primary spermatocytes, and round spermatids (Table 1, Figure 2).

SIM was used to identify the precise localization of αV integrin in individual membrane compartments of epididymal acrosome-intact mouse sperm. SIM findings revealed that αV integrin is mainly localized over the acrosomal cap region of acrosome-intact mouse sperm heads, and it was specifically defined on the plasma membrane (PM) strongly covering the apical hook, and lightly covering the post-acrosomal region and neck (Figure 3a–c). In the acrosomal cap area, αV integrin is likely to be expressed in the PM and both outer acrosomal (OAM) and inner acrosomal (IAM) membranes. Changes in αV integrin localization during the acrosome reaction in capacitated sperm (Appendix A) were detected by confocal microscopy (Figure 3d–f). After the acrosome reaction, the most prominent signal for αV integrin was captured in the IAM defining the equatorial segment with the remaining strongly labeled hook, and a weak expression was also detected in the post-acrosomal region of the sperm head. There was a particular enrichment of the signal in the sperm neck and tail.

Western blot analysis was performed using protein extracts from epididymal (Ep) and acrosome-reacted (AR) sperm, and 130 kDa protein was detected by immunodetection on the PVDF membrane (Figure 3g). We compared protein abundance between the expression of αV integrin in samples of epididymal and acrosome-reacted sperm by densitometric analysis from three measurements and detected that the signal of protein αV integrin was normalized to the loading control of tubulin in the samples (Figure 3g,h). The graph (Figure 3h) shows a decreased amount of αV integrin in acrosome-reacted sperm, and the decrease is significant (*p* < 0.05; *). Using Western blot immunodetection, αV integrin was also determined as a 130 kDa protein in mouse testicular (T) and epididymal (E) tissues. Apart from 130 kDa protein, we observed a signal in a band with a molecular mass of 60 kDa, which is non-specific according to the datasheet for the antibody (Figure 3g).

### 2.2. Porcine Sperm

αV integrin was detected in the acrosomal cap region and in the neck of ejaculated acrosome-intact porcine sperm. Based on the images obtained by SIM, the αV integrin localization was mainly defined to the PM and both acrosomal membranes enriched in OAM overlaying the apical acrosomal area and PM covering the neck region (Figure 4a–c). During the acrosome reaction of capacitated sperm (Appendix A), the localization of αV integrin moved (Appendix A) across the whole acrosome-reacted sperm head and was observed in IAM and the PM covering the equatorial segment, post-acrosomal region, and enriched in the neck region when compared with acrosome-intact sperm; however, the limits of the confocal method need to be considered (Figure 4d–f). Light staining was also observed in the sperm tail but without a specific change during the acrosome reaction. To confirm the αV integrin localization in distinct sperm membranes structures, whole sperm membrane fractionation was performed to obtain a protein pool comprised of the PM, OAM and IAM. Using selected membrane fractions, the αV integrin (130 kDa) was positively detected in IAM and the PM (Figure 4g). PM detection most probably originated from the PM covering the neck region, which agrees with immunofluorescent labeling (Figure 4a–c). αV integrin was also positively detected on the PVDF membrane in the extract from ejaculated acrosome-intact (Ej) sperm and acrosome-reacted (AR) sperm (Figure 4g). Detected signals were analyzed by densitometric analysis, and the samples were normalized to tubulin. The graph (Figure 4h) reflects a decreased intensity of antibody reaction to αV integrin in the acrosome-reacted sperm (*p* > 0.5). The presence of αV integrin (130 kDa) was proven by Western blot analysis in protein extracts from testicular (T) and epididymal (E) tissues (Figure 4g).

### 2.3. Human Sperm

Human sperm and porcine sperm have a very similar morphology, which makes porcine sperm a good model to study human reproduction. However, there are still interspecies differences in sperm protein expression and behavior prior to fertilization; therefore, targeted comparison is needed. SIM was employed to address the precise localization of αV integrin in ejaculated acrosome-intact human sperm (Figure 5a–c). Immunofluorescent labeling showed that αV integrin is localized over the PM surrounding the sperm head in human ejaculated acrosome-intact sperm with the strongest signal in the acrosomal cap area, especially in the equatorial segment. Both acrosomal membranes (OAM and IAM) were detectable with individual Z-stack SIM evaluation. After the acrosome reaction of capacitated sperm (Appendix A), the staining pattern dramatically changed (Figure 5d–f). The αV integrin localization shown by confocal microscopy remained on the IAM and over the equatorial segment, with strong enrichment in the PM overlaying the post-acrosomal region, neck and sperm tail. Protein immunodetection from acrosome-intact sperm on the PVDF membrane provided a specific signal corresponding to αV integrin (130 kDa) in the PM and lightly in the IAM fraction. There was also a non-specific signal (60 kDa) detected in the IAM fraction (Figure 5g).

Due to the heterogeneity of human sperm within an ejaculate, we were receiving highly diverse readings for the abundance of αV integrin protein between capacitated samples after the induction of the acrosome reaction (CaI; Appendix A). For this reason, we decided to separate acrosome-intact and acrosome-reacted sperm into individual fractions based on acrosome integrity, which was monitored by PNA staining (Appendix A). In the selected fraction strongly enriched by acrosome-reacted (AR) sperm, we immunodetected a decreased (*p* < 0.02; **) amount of αV integrin (130 kDa) compared to the samples of ejaculated (Ej) sperm (Figure 5h). Similar to the mouse and porcine sperm, we detected αV integrin (130 kDa) in protein extracts from the testicular (T) and epididymal (E) tissues (Figure 5g).

## 3. Discussion

Integrins have been confirmed to be involved in mammalian reproductive processes [17] including those of human. In sperm, they are involved in both sperm maturation and sperm–egg interaction [2]. αV integrin is one of the members of the integrin family, and its presence is described on oocytes [18,19,20,21] and sperm [10,22,23]. Nevertheless, questions about gene activity in the germ cell subsets during spermatogenesis as well as protein expression in individual sperm head membrane compartments, and changes during membrane raft reorganization during the sperm acrosome reaction, remain unanswered. Furthermore, interspecies differences related to distinct sperm morphology and fertilization strategies play a key role in protein network organization and were identified in other proteins of the integrin family [16,24,25,26]. They were also identified in tetraspanins and izumo1 or CD46 proteins [8,9,26,27]; when even timing of re-localization reflects individual species differences [26,28,29], including the absence of the protein itself [30,31].

Within this study, we address all the above aspects in detail and delivered comprehensive data regarding *αV integrin* gene expression in the male germ cell subpopulation during spermatogenesis with extended αV integrin protein comparison among three species with distinct sperm head morphology and fertilization behavior. Unlike studies referring to the localization of αV integrin on the IAM [10,14], we detected αV integrin on the PM after performing the fractionation of human and porcine sperm. This finding was confirmed by data generated from SIM, and it was specified that αV integrin localization is within the PM overlaying the acrosome region, neck, and tail on sperm for all three species. The αV integrin membrane expression in these three membrane compartments may reflect its adhesive function to the oviductal epithelium accompanied with signalling communication [32].

The acrosome reaction is a complex event, which involves massive membrane restructure and protein relocation [33]. Membrane rafts including the expressed proteins are repositioned and either released during the acrosome vesicle exocytosis or reformed to compartments playing a key role during sperm–oocyte membrane fusion [34]. We traced the destiny of αV integrin during this process and observed its decrease in acrosome-reacted sperm. The decrease most likely reflects protein release during parting of the PM and OAM vesicles as an essential step of the acrosome reaction. This finding complements immunofluorescent labeling of acrosome-reacted sperm, where αV integrin is confined to the IAM. At this point, our findings are fully in agreement with the previously reported localization of αV integrin in mouse [10] and human [14] sperm, and it can possibly reflect sperm with lost or damaged acrosomes or pathological sperm with acrosome impairment. With these results, and with the knowledge that αV integrin abundance and localization changes when the acrosome is absent, this protein could be a marker of sperm quality. Importantly, the correlative results using protein extract and super-resolution imaging of protein localization show the necessity of detailed protein capturing. Based on the SIM findings, it was discovered that αV integrin is also expressed in the PM of the equatorial segment, where the IAM merges with the PM that extends over the post-acrosomal segment. Therefore, protein extract analysis results in a certain bias because the IAM fraction is likely to contain a small amount of protein of PM origin. In agreement with the nature of lipids rafts, of which αV integrin is part [35], and due to a change in membrane fluidity during capacitation, the lateral movement of αV integrin in the membrane is likely to occur, resulting in the presented protein re-localization.

The presence of αV integrin on the sperm surface can have several important roles during fertilization. One of them is the fusion of oviductosomes with the sperm, wherein the interaction is mediated in sperm by integrin heterodimers formed with a αV integrin subunit [13]. Considering that αV integrin is involved in fusion exosomes, and we detected αV integrin on the surface of acrosome-reacted sperm heads in all three species, αV integrin may be part of the interaction and fusion of sperm with the oocyte. This hypothesis is also supported by the fact that αV integrin recognizes peptides with the RGD sequence, which is part of the EC2 domain on tetraspanins CD9, CD81 and CD51 present on oocytes of different species [9,11,12,36]. The interaction of gametes mediated by αV integrin were studied using peptides with the RGD sequence, which had an inhibitory effect on the adhesion and oocyte penetration [19,22]. On the other hand, the inhibitory effect was not achieved by using the antibodies against αV integrin [37], but the incubation of anti-αV integrin antibodies with sperm inhibited the effect on gamete fusion [10].

## 4. Materials and Methods

Unless otherwise noted, all chemicals were purchased from Sigma-Aldrich (St. Louis, MO, USA).

### 4.1. Animals

Inbred C57BL/6 mice were housed in a breeding colony of the Laboratory of Reproduction, IMG animal facilities, Institute of Molecular Genetics of Czech Academy of Science, and food and water were supplied ad libitum. The male mice used for all experiments were healthy, 10–12 weeks old, with no sign of stress or discomfort. All animal procedures and experimental protocols were approved by the Animal Welfare Committee of the Czech Academy of Sciences, Animal Ethics protocol code 66866/2015-MZE-17214, 18 December 2015.

### 4.2. Tissue Preparation

The whole mouse *cauda epididymis* was dissected into smaller pieces and placed into the tube and homogenized. The mouse *testis* was cut in half, and one half was placed into the tube for following homogenization. Tissues of the *testis* and *cauda epididymis* of slaughtered pigs were cut into small pieces and kept at −70 °C for the protein extraction. Then, 10 mg of mouse or porcine tissue of *testis* or *epididymis* was placed in the Precellys tubes with beads (Bertin Technologies, Fontaine, France) and homogenized in 500 µL lysis buffer (50 mM Tris.HCl, pH 7,8, 1%Triton, and 30 mM KCl) by a Precellys tissue homogenizer (5000 rpm, 10 s, 3 times, 4 °C; Bertin Technologies). Consequently, the samples were centrifuged at 10,000× *g*, for 10 min, at 4 °C, and the supernatant was transferred into a new tube and was used for SDS-PAGE. Lysate from adult human *testis* (ab30257-10) and *epididymis* (ab29975-1) were from Abcam (Cambridge, UK).

### 4.3. Elutriation

A centrifuge J26XP with an elutriation rotor JE-5.0 (Beckman Coulter, Indianapolis, IN, USA) were used for obtaining the elutriation fraction. The elutriation protocol was carried out in PBS at 4 °C; the protocol is described in [16]. The cells gained in each tube were pelleted by centrifugation (400× *g*, 15 min, 4 °C) and resuspended in TRI Reagent^®^. The total RNA was isolated according to manufacturer’s instructions and stored at −70 °C.

### 4.4. Reverse Transcription and Real Time Quantitative PCR (qRT-PCR)

Total RNA was isolated from testicular fractions prepared by elutriation and *testis* samples using TRI Reagent^®^. RNA extracts (2 μg) were treated with DNase I (1 U/µL, Fermentas, Hanover, MA, USA) in the presence of DNase I buffer 10 × (Thermo Scientific, Waltham, MA, USA) with MgCl_2_ for 30 min at 37 °C, and EDTA (Fermentas) was added for 10 min at 65 °C. The reverse transcription reaction contained 5 × reaction Buffer (Fermentas), Riboblock Inhibitor (20 U/µL), Universal RNA Spike II (0.005 ng/µL, TATAA biocenter, Göteborg, Sweden), 10 mM dNTP Mix (Thermo Scientific), oligo(dt)18 (Thermo Scientific) mixed 1:1 with Random primers (Thermo Scientific) and M-MuLV RevertAid transcriptase (200 U/µL, Fermentas), and run to generate cDNA. All cDNA samples were synthesized in duplicates. The RT^-^ negative control was prepared in the same conditions but with RNase/DNase-free water. For qRT-PCR, 10 ng/µL cDNA was used. Two times Maxima SYBR Green qPCR Master Mix (Thermo Scientific), reverse and forward primer (1 µM, Generi Biotech, Hradec Kralove, Czech Republic), and nuclease-free water were used, and all reactions were performed in duplets in a PCR cycler (CFX 384-qPCR cycler, Bio-Rad). The Ribosomal protein S2 (Rps2) gene was used as the reference gene. Specific gene markers for germinal cells and somatic cells were used to determine elutriation fractions. The RT^-^ negative control for cDNA synthesis was also analyzed.

### 4.5. Sperm Preparation

Both *caudae epididymides* were dissected, and sperm from their distal regions were released into a two 200 μL droplets of M2-fertilizing medium (M7167) under paraffin oil (P14501, P-LAB, Prague, Czech Republic) in a Petri dish and pre-tempered at 37 °C in the presence of 5% CO_2_. Released sperm were assessed for motility and viability under a light inverted microscope with a thermostatically controlled stage at 37 °C with the Computer-Assisted Sperm Analysis (CASA) module and software (Proiser & Zoitech, Paterna, Spain). In contrary to pig and human, it is not possible to use the ejaculated sperm in the case of mouse. However, in our study, the αV transmembrane protein was investigated, and we do not presume any dramatic changes in this protein behavior during ejaculation.

Porcine spermatozoa were obtained from Insemination Station Skrsin (NATURAL, s.r.o., Prague, Czech Republic). Porcine ejaculated spermatozoa were centrifuged from extender medium at 300× *g* for 10 min at room temperature and were washed twice in PBS, centrifugation 300× *g* for 10 min at room temperature.

Human ejaculates were obtained from men after 3–4 days of sexual abstinence, at the Centres for Assisted Reproduction (Prague, Czech Republic) with the informed consent of the healthy donors and in accordance with the approval of the Ethics Committee, protocol code BIOCEV 012019, 20 January 2019. After liquefaction, the human sperm were separated from seminal plasma by centrifugation gradient (55%/80%), SupraSperm^®^ System (Origio, Måløv, Denmark) and centrifuged at 300× *g* for 20 min at 37 °C. Biological materials and experimental protocols were approved by the Ethics Committee of the General University Hospital (Prague, Czech Republic), protocol code 617/17 S-IV.

### 4.6. In Vitro Capacitation and Acrosome Reaction Induction

Epididymal mouse sperm (5 × 10^6^/mL) were capacitated in 100 µL M2 medium (M7167) for 90 min at 37 °C in 5% CO_2_. Capacitated status of mouse sperm was determined by staining of anti-phosphotyrosine antibody (PTyr 01 112630025, Exbio Antibodies, Prague, Czech Republic) [38]. The acrosome reaction was induced by 5 µM Calcium ionophore (CaI, A23187) for 90 min at 37 °C in 5% CO_2_.

Capacitated medium for porcine sperm was prepared from TALP-stock medium (20 mM HEPES; 114.06 mM NaCl; 25.07 mM NaHCO_3_; 3.2 mM KCl; 0.35 mM NaH_2_PO_4_.H_2_O; 0.5 mM MgCl_2_.6H_2_O; 8 mM lactate Ca.5H_2_O; 10 mM sodium lactate; 5 mM glucose; 2 mM caffeine; 0.17 mM kanamycin sulphate, and 1 mg/mL polyvinyl alcohol). Before using the TALP-stock medium, it was supplemented with 3 mg/mL bovine serum albumin (BSA) and 0.12 mg/mL sodium pyruvate, and the pH was adjusted to 7.4. The porcine sperm (5 × 10^6^/mL) were incubated in capacitated medium for 2 h at 37 °C in 5% CO_2_. The capacitated status of porcine sperm was determined by staining of anti-phosphotyrosine antibody (4G10, Merck Millipore, Burlington, MA, USA) according to [39]. The acrosome reaction was induced by 5 µM CaI for 1 h at 37 °C in 5% CO_2_.

Human ejaculated sperm (5 × 10^6^/mL) were capacitated in sperm preparation medium (Origio) for 2 h at 37 °C in 5% CO_2_. The capacitated status of human sperm was determined by staining of anti-phosphotyrosine antibody (4G10, Merck Millipore) according to [40]. Subsequently, 5 µM CaI was added to the medium for 1 h at 37 °C in 5% CO_2_ to induce the acrosome reaction.

### 4.7. Immunofluorescent Detection of αV integrin with Confocal Microscopy and Structured Illumination Microscopy

Freshly released epididymal mouse sperm were used for confocal microscopy and SIM. Prepared epididymal mouse sperm and ejaculated human and porcine sperm were washed twice in PBS, smeared onto glass slides, and air-dried. For SIM, sperm samples were always prepared onto high-precision cover glasses (thickness No. 1.5 H, 170 ± 5 μM, Paul Marienfeld GmbH and Co. KG, Lauda-Königshofen, Germany). Sperm smears were fixed with 2% paraformaldehyde (PFA) for 10 min, which was followed by washing in PBS. After fixation, the sperm were permeabilized by incubation in Intracellular Staining Perm Wash Buffer (Biolegend, San Diego, CA, USA) 3 times for 5 min. Sperm were blocked with 5% BSA in PBS for 1 h and incubated with primary antibody (anti-integrin αV antibody, AB1930) overnight at 4 °C, which was followed by secondary Goat-anti rabbit antibody (Alexa Fluor 488, A11008; Invitrogen, Waltham, MA, USA) 1:300 in PBS for 1 h at room temperature. After the application of the primary and secondary antibodies, sperm were incubated with PNA lectin, Alexa Fluor 568 (L32458; Invitrogen), diluted 1:500 for 30 min, and consequently incubated with DAPI (0.85 μg/mL, Thermo Scientific) for 5 min and washed in PBS. At the end, sperm were washed in distilled water and air-dried. Dry samples were covered with 90% glycerol with 5% anti-fade N-propyl gallate. Multi-color SIM super-resolution images were obtained by DeltaVision OMX™ V4 (Light Microscopy Core Facility, IMG CAS, Prague, Czech Republic). An open-source software Fiji [41] was used for further image processing.

### 4.8. Membrane Fractionation

Isolation of sperm plasma membrane and acrosomal membrane was made according to [42]. This methodology is demanding on sperm amount, and therefore, it could have been performed only with porcine and human sperm. Fresh ejaculated porcine or human sperm were prepared as above and diluted 1:2 with Krebbs Ringer Bicarbonate medium: 0.7 mM Na_2_HPO_4_; 0.49 mM MgCl_2_; 4.56 mM KCl; 119.8 mM NaCl; 1.3 mM NaH_2_PO_4_; 2.37 mM fructose; 14.9 mM NaHCO_3_ (pH 7.0–7.6). Diluted semen of mouse and human were layered on solution of 1.3 M sucrose with 0.9% NaCl and centrifuged for 30 min at 2000× *g* at 4 °C. Pellet of sperm was resuspended in 0.15 M NaCl with 5 mM HEPES (pH 7.0), layered on solution of 1.3 M sucrose with 0.9% NaCl, and centrifuged for 20 min at 34,000× *g* at 4 °C. The sperm pellets were resuspended in 0.15 M NaCl with 5 mM HEPES (pH 7.0) supplemented with protease inhibitors and sonicated (Hielscher Ultrasonics GmbH, Teltow, Germany) 1 × 5 s and 3 × 2 s for mouse, 10 × 10 s intervals for porcine, and 20 × 10 s intervals for human. Membranes from homogenate were separated by a discontinuous sucrose gradient consisting of 1.75 M sucrose and 1.3 M sucrose (1:1) and centrifugation for 3 h at 95,000× *g* at 4 °C. The plasma membrane (PM) fraction was at the interface between the sample and 1.3 M sucrose solution; the outer acrosomal membrane (OAM) fraction was the interface between 1.3 M/1.75 M sucrose. The pellet contained an inner acrosomal membrane (IAM), and the remaining equatorial segments were closely associated with the sperm heads. The fractions of PM and OAM were diluted with PBS and pelleted by centrifugation for 30 min at 120,000× *g* at 4 °C. The fraction of IAM was washed by PBS and centrifuged at 3500× *g* for 10 min at 4 °C. Subsequently, the separate membrane fractions were solubilized with 1% (*v/v*) Triton X-100 for 1 h at 4 °C. Proteins were precipitated by acetone, and then, samples were incubated in reducing sample buffer for SDS-PAGE at 100 °C for 5 min. A large number of mice would be needed in order to collect the required amounts of epididymal sperm for the membrane fractionation; for ethical reasons, this experiment was not performed.

### 4.9. Selection of Human Sperm

After the acrosome reaction, sperm were washed in PBS, and 5 × 10^6^ sperm were fixated in a mixture of frozen acetone–methanol (1:1). After washing, the sperm pellet was diluted in 80 µL of PBS. The sperm suspension was incubated with PNA lectin, Alexa Fluor 568 (L32458; Invitrogen); then, it was diluted 1:500 for 30 min and subsequently with DAPI (0.85 μg/mL, Thermo Scientific) for 5 min. Sperm were sorted by BD FACSDiva™ (BD Biosciences, Franklin Lakes, NJ, USA) into two fractions: acrosome-intact stained by PNA lectin in the acrosome and acrosome-reacted with PNA stained only in the equatorial segment. Sperm were washed with PBS and centrifuged at 500× *g* for 5 min. Sperm pellets were dissolved in nonreducing sample buffer for 5 min at 95 °C.

### 4.10. SDS Electrophoresis and Western Blot Analysis

A pellet of mouse (epididymal and acrosome-reacted), porcine, and human (ejaculated and acrosome-reacted) sperm were dissolved in nonreducing sample solutions with protease inhibitors for 30 min at 4 °C and subsequently boiled for 5 min at 95 °C. The protein extracts from mouse, porcine, and human sperm were separated by 10% SDS-PAGE, and sperm membrane fractions were separated by 12% SDS-PAGE. The proteins from sperm and sperm membrane fractions were transferred onto the PVDF membrane Immobilon^®^-P (Merck Millipore). The molecular weight of the proteins was assigned by Precision Plus Protein Dual Color Standards (Bio-Rad, Hercules, CA, USA). The membranes were blocked in 5% Blotting-Grade Blocker (Bio-Rad) and incubated with rabbit monoclonal recombinant anti-integrin αV antibody (ab179475, Abcam) diluted 1:500 in PBS at 4 °C overnight. After incubation, the membranes were washed in PBS and incubated with secondary antibody, goat anti-rabbit IgG conjugated to horseradish peroxidase (Bio-Rad), diluted 1:3000 in PBS for 1 h at RT, and consequently washed in PBS. The membranes were developed with SuperSignal™ Chemiluminescent Substrate (Thermo Scientific). The membrane incubated with rabbit IgG (Invitrogen) with a concentration of 4 μg/mL was used as a negative control.

### 4.11. Statistical Analysis

For all three independent replicates, immunodetection of transferred proteins was performed. Each data point is presented as a mean ± SD. An independent two-sample *t*-test in GraphPad Prism 5 software (GraphPad Prism Software, Inc., La Jolla, CA, USA) was used for statistical evaluation of densitometric analysis.

## 5. Conclusions

Delivering positive gene expression of *αV integrin* in the individual germ cell subpopulation during spermatogenesis was followed by its protein expression and membrane localization on mouse, porcine, and human reproductive tissue, sperm. Depiction of the exact localization of αV integrin on the sperm head and its novel compartmentalization after the acrosome reaction may presume that αV integrin is a molecule involved in sperm adhesion, interaction, communication with the oviduct, and gamete membrane fusion. The findings relevant to human provide grounds for further research into fertility assessment using αV integrin as a marker for reflecting sperm quality.

## Figures and Tables

**Figure 1 ijms-22-09525-f001:**
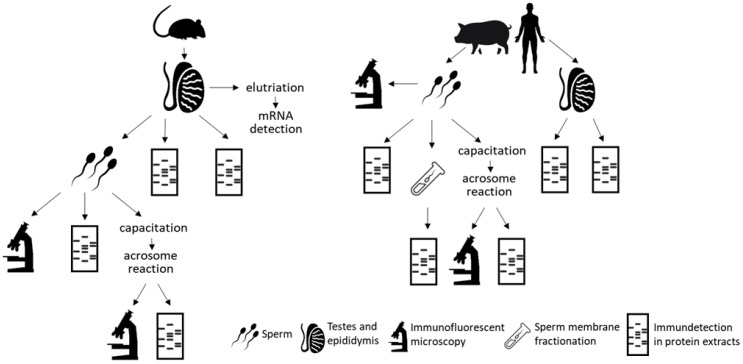
Scheme describing design of experimental study and use of selected methods.

**Figure 2 ijms-22-09525-f002:**
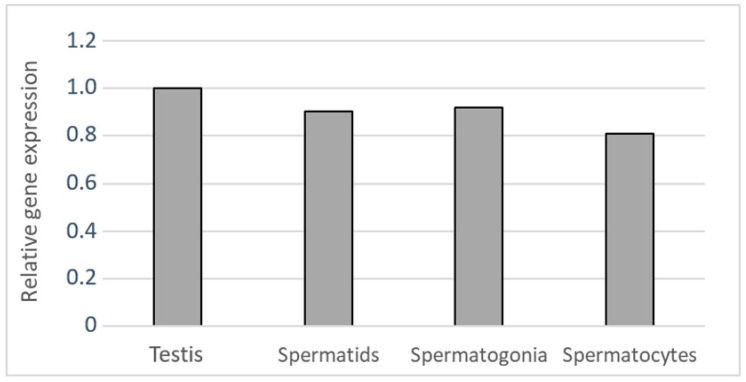
Gene expression of *αV integrin* in the mouse testicular elutriation fractions enriched with round spermatids (Spermatids), spermatogonia, and primary spermatocytes (Spermatocytes).

**Figure 3 ijms-22-09525-f003:**
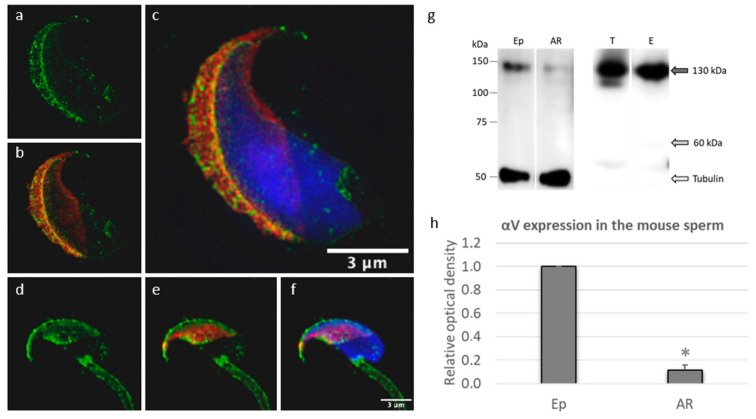
Detection of αV integrin in mouse spermatozoa and reproductive tissues. (**a**–**c**) The αV integrin (green) localization on the intact sperm head was visualized by SIM and confined to the PM defining the acrosomal region, extending to the apical hook region with a light signal over the post-acrosomal region and neck. Both OAM and IAM were detectable. (**d**–**f**) The αV integrin (green) localization after the acrosome reaction was captured by confocal microscopy and depicted strongly in IAM covering the equatorial segment including the PM extending to the post-acrosomal region enriched in the neck and tail. The PM of the apical hook region remained strongly labeled. Acrosomal status was visualized by PNA staining (red); Hoechst (blue) was used for detection of the nucleus. Scale bars represent 3 µM. (**g**) Western blot immunodetection of αV integrin in the extracts from epididymal sperm (Ep), acrosome-reacted sperm (AR), *testis* (T), and *epididymis* (E). Anti αV integrin antibody detected bands with a molecular weight of 130 kDa (gray arrow). (**h**) Western blot densitometry analysis in protein extracts normalized to the amount of tubulin shows a decreased expression of αV integrin (*p* < 0.05; *) in acrosome-reacted (AR) sperm in comparison with epididymal (Ep) sperm.

**Figure 4 ijms-22-09525-f004:**
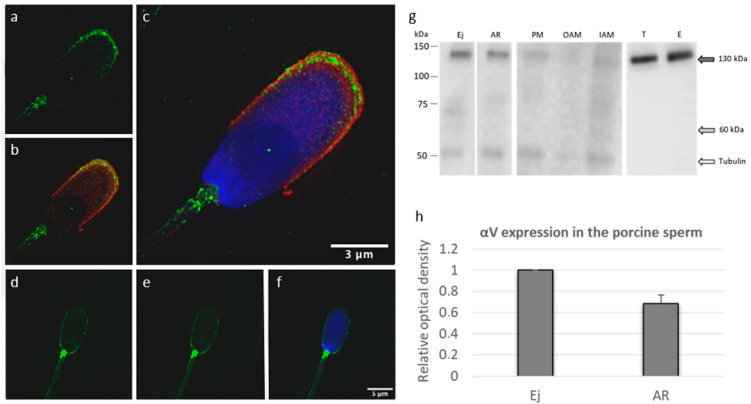
Detection of αV integrin in porcine sperm and reproductive tissues. (**a**–**c**) The αV integrin (green) localization on acrosome-intact ejaculated sperm was visualized by SIM and confined to the PM covering the acrosomal region and acrosomal membranes with enrichment in OAM. (**d**–**f**) The αV integrin (green) localization after the acrosome reaction was captured by confocal microscopy and observed in IAM and the PM covering the post-acrosomal segment, with neck enrichment. Acrosomal status was visualized with PNA staining (red); Hoechst (blue) was used for detection of the nucleus. Scale bars represent 3 µM (**a**–**c**) and 5 µM (**d**–**f**). (**g**) Western blot immunodetection of αV integrin using protein extracts from the ejaculated sperm (Ej), acrosome-reacted sperm (AR), PM, OAM, and IAM fractions from ejaculated sperm, *testis* (T), and *epididymis* (E). Antibody detected bands show protein of 130 kDa size (gray arrow). (**h**) Western blot densitometry analysis of αV integrin expression in ejaculated (Ej) and acrosome-reacted (AR) sperm normalized to the amount of tubulin shows a visible αV integrin decrease (*p* > 0.05) in AR sperm.

**Figure 5 ijms-22-09525-f005:**
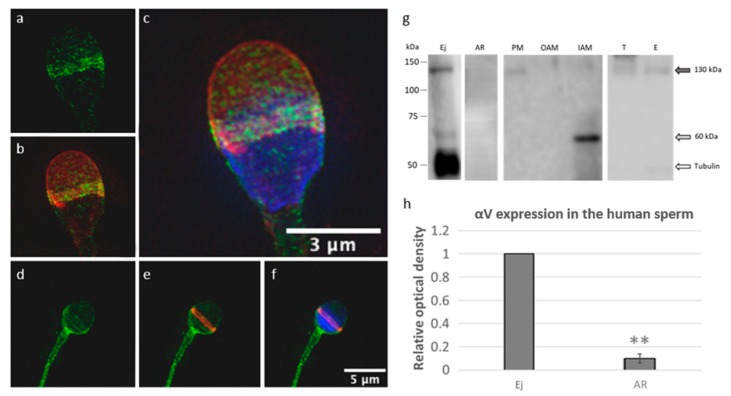
Detection of αV integrin in human spermatozoa and reproductive tissue. (**a**–**c**) The αV integrin (green) localization on acrosome-intact ejaculated sperm was visualized by SIM and confined to the PM, OAM, and IAM overlaying the apical acrosome and equatorial segment. (**d**–**f**) The αV integrin (green) localization after the acrosome reaction was captured by confocal microscopy and depicted in IAM and the PM overlaying the equatorial segment, post-acrosomal region, neck, and tail. Acrosomal status was visualized with PNA staining (red); Hoechst (blue) was used for detection of nucleus. Scale bars represent 3 µM (**a**–**c**) and 5 µM (**d**–**f**). (**g**) Western blot immunodetection of αV integrin in extracts from ejaculated sperm (Ej), acrosome-reacted sperm (AR), PM, OAM, and IAM fractions from ejaculated sperm, *testis* (T), and *epididymis* (E). Antibody detected bands with a molecular weight of 130 kDa (gray arrow). (**h**) Western blot densitometry analysis in protein extracts shows a decreased expression of αV integrin (*p* < 0.02; **) in acrosome-reacted (AR) sperm in comparison with the ejaculated (Ej) sperm.

**Table 1 ijms-22-09525-t001:** Gene expression of *αV integrin* in the cell-type fractions. Cq value of the gene is normalized by reference gene *Rps2*.

Target Gene	Testicular Elutriation Fractions			
	Testis	Round Spermatids	Spermatogonia	Primary Spermatocytes
*αV integrin*	1	0.904	0.921	0.808

## Data Availability

The study did not report any data.

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
