# Peer review of "αV Integrin Expression and Localization in Male Germ Cells"

_ijms, 2021, doi:10.3390/ijms22179525_

Round 1
Reviewer 1 Report
Review Manuscript ID: ijms-1354875, entitled „αV integrin expression and localization in male germ cells”
The aim of the present study was to determine the exact location of the αV integrin subunit in intact sperm heads and to investigate its potential changes in membrane reorganization during the acrosomic reaction initiated in the captured sperm.
The peer-reviewed manuscript is interesting. Nevertheless, I have few points, which in my opinion should be explained:
L 278 How were the testes and epididymis tissues collected, and which parts of them were used for research? Introduce the methodology
L 310 Was the sperm collected from only one epididymis or both? Explain this.
L 313 Describe how motility and viability was assessed
L 431 The statement 'makes αV integrin a molecule involved in sperm adhesion, inter action, communication with the oviduct, and gamete membrane fusion' is a pretty bold statement. The authors did not examine this effect of αV integrin.
Author Response
Thank you very much for valuable points to our manuscript, we addressed them all. Please see the response to your points below. They were also incorporated in the revised paper.
L 278 How were the testes and epididymis tissues collected, and which parts of them were used for research? Introduce the methodology
Answer: In case of mouse, the whole epididymis was used. The epididymis was dissected into smaller pieces, placed into tube and homogenized. The testis was halved and one part was used for homogenization. We added this information into the manuscript. Please see lines 278-280.
Porcine testis and epididymis were obtained from slaughtered pigs in slaughterhouse. After transferring of tissue to the lab, the testis and cauda epididymis were cut into pieces of size approximately 0,5 x 0,5 cm and frozen in -70°C for further use. We added the information into the manuscript. Please see lines 280-282.
Unfortunately for human we are not able to answer the question because for experiments we used the lysate from adult human testes (ab30257-10) and epididymis (ab29975-1), that is provided by Abcam (Cambridge, UK) as mentioned in methodology. Please see lines 288-289.
L 310 Was the sperm collected from only one epididymis or both? Explain this.
Answer: Sperm were collected from both epididymis of mice. Portion of them was used for immunofluorescent experiments and portion for detection of αV in protein extracts. Please see lines 318. .
L 313 Describe how motility and viability was assessed
Answer: We evaluated motility and viability of mouse sperm by CASA. We added this fact in the Methods. Please see lines 323-324.
L 431 The statement 'makes αV integrin a molecule involved in sperm adhesion, interaction, communication with the oviduct, and gamete membrane fusion' is a pretty bold statement. The authors did not examine this effect of αV integrin.
Answer: Based on knowledge of other authors referred in the discussion, and our findings we suppose a similar role of αV integrin to other integrins. We accept that our statement in the conclusion is too bold. We rewrote the sentence.
Depiction of exact localization of αV integrin on the sperm head and its novel compartmentalization after the acrosome reaction may presume that αV integrin is a molecule involved in sperm adhesion, interaction, and gamete membrane fusion.
Please see line 446.
Reviewer 2 Report
The study deals with a highly up-to-date and interesting topic, providing valuable information to the readers interested in andrology. The mansucript is very complex however it reads well and presents with data that have the potential to be explored further.
My two suggestions would be to add limitations that the authors had to deal with during the experimental execution. Also, because the methodology is quite complex, taking advantage of a variety of samples/species/techniques, I would suggest to add a simple scheme to add more clarity to all the experiments done in the study.
Author Response
Thank you very much for your valuable points which helped us to clarify the information provided in the revisesed paper.
In our manuscript, we present information about behaviour of αV integrin in sperm of mouse, pig and human. As you mentioned using of three completely different mammalian species and wide range of used methods exposed our work to some of the limitations we had to deal it with.
Based on your suggetiongs we prepared a scheme describing design of experimental study and use of selected methods and placed it as a Figure 1.
The limitations of the study we included into relevant methodological sections, which summarise nicely all the experimental models. Please see the text below and lines: 324-327, and 405-407.
In contrary to pig and human, it is not possible to use the ejaculated sperm in the case of mouse. However, in our study the αV transmembrane protein was investigated and we don’t presume any dramatic changes in this protein behaviour during ejaculation. A large number of mice would be needed in order to collect required amount of epididymal sperm for the membrane fractionation, for ethical reasons this experiment was not performed.